# Disentangling Factors of Variations Using Few Labels

Francesco Locatello[2,3], Michael Tschannen[2], Stefan Bauer[3], Gunnar Rätsch[2], Bernhard Schölkopf[3], and Olivier Bachem[1]

[1]Google Research, Brain Team
[2]ETH Zurich
[3]Max-Planck Institute for Intelligent Systems

## Abstract

Learning disentangled representations is considered a promising research direction in representation learning. Recently, Locatello et al. (2018) demonstrated that the unsupervised learning of disentangled representations is theoretically impossible and that state-of-the-art methods, which are often unsupervised, require access to annotated examples to select good model runs. Yet, if we assume access to labels for model selection, it is not clear why we should not use them directly for training. In this paper, we first show that model selection using few labels is feasible. Then, as a proof-of-concept, we consider a simple semi-supervised method that directly uses the labels for training. We train more than 7000 models and empirically validate that collecting a handful of potentially noisy labels is sufficient to learn disentangled representations.

## 1 Introduction

Learning the true underlying generative model for the observed data is one of the fundamental problems of machine learning. It is commonly accepted that the observed data $\mathbf{x}$ exhibits redundant features and its essence can be captured by a low dimensional latent variable $\mathbf{z}$ (Bengio et al., 2013; Kulkarni et al., 2015; Chen et al., 2016; Tschannen et al., 2018). The observations are generated by first sampling a $\mathbf{z}$ from a distribution $P(\mathbf{z})$. Then, $\mathbf{x}$ is sampled from the conditional distribution $P(\mathbf{x}|\mathbf{z})$. The goal of disentanglement learning is to find a transformation of the data $r(\mathbf{x})$ which is "aligned" with the factors of variations in $\mathbf{z}$. The hope is that such representation will be useful for any downstream task (Bengio et al., 2013; Peters et al., 2017; LeCun et al., 2015; Bengio et al., 2007; Schmidhuber, 1992; Lake et al., 2017; Goodfellow et al., 2009; Lenc & Vedaldi, 2015; Tschannen et al., 2018; Higgins et al., 2018; Suter et al., 2018). In particular, we hope to learn a disentangled representation without supervision so that only few labels will be needed to learn any downstream task (Bengio et al., 2013; Schölkopf et al., 2012; Peters et al., 2017; Pearl, 2009; Spirtes et al., 1993).

Current state-of-the-art approaches enrich the *Variational Autoencoders (VAEs)* (Kingma & Welling, 2013) objective with some unsupervised regularizer that should encourage disentangled representations (Higgins et al., 2016; Burgess et al., 2018; Kim & Mnih, 2018; Chen et al., 2018; Kumar et al., 2017; Rubenstein et al., 2018). Unfortunately, purely unsupervised disentanglement learning is theoretically impossible and unsupervised methods exhibit a considerable variance across different runs (Locatello et al., 2018). Therefore, labels (which in this setup are the observations of the factors of variations) or human supervision are in practice required to perform model selection.

In this paper, we investigate how much supervision is needed to successfully perform model selection on the unsupervised methods and how does it compare to a simple semi-supervised approach. For this purpose, we re-evaluate the $12\,000$ trained models of Locatello et al. (2018) using 100 and 1000 labelled examples. We further train more than 7000 simple semi-supervised models using the same amount of labels in a 90%/10% train/validation split.

Overall, we show that a suprirsing little amount of supervision is enough to learn disentangled representations. Our contributions can be summarized as follows: (i) we show that disentanglement

metrics can be used to tune the hyperparameters of unsupervised methods even if very few labelled examples are available. (ii) we show that adding a simple supervised loss to $\beta$-VAE outperforms unsupervised training with supervised model selection using as little as 100 labelled examples. (iii) we show that the simple semi-supervised method we consider is robust to label noise and simplified labels.

## 2 INCORPORATING LABEL INFORMATION

Let $R_u(q_\phi(\mathbf{z}|\mathbf{x}))$ be a regularizer imposing structure on the encoder distribution in order to encourage a disentangled representation. Then, we can group all the SoTA unsupervised approaches (Higgins et al., 2016; Burgess et al., 2018; Kim & Mnih, 2018; Chen et al., 2018; Kumar et al., 2017) under the following optimization template:

$$\max_{\phi,\theta} \quad \mathbb{E}_{\mathbf{x}}[\mathbb{E}_{q_\phi(\mathbf{z}|\mathbf{x})}[\log p_\theta(\mathbf{x}|\mathbf{z})] - D_{\mathrm{KL}}(q_\phi(\mathbf{z}|\mathbf{x})\|p(\mathbf{z})) + \beta R_u(q_\phi(\mathbf{z}|\mathbf{x}))] \tag{1}$$

In practice, even though disentangled methods might train without supervision, it is assumed that some number of observations of $\mathbf{z}$ is available for evaluation. Note that supervised model selection explicitly breaks the impossibility result of (Locatello et al., 2018). While collecting a large amount of labelled data in this setup is unreasonable, we question whether labelling a very limited amount of data points is sufficient to reliably obtain disentangled representations.

If we collect a large dataset of unlabelled data and a few labelled examples we have two options: either we train an unsupervised method and perform supervised model selection or we split the labels in two sets and use one training and one for validation.

As a proof-of-concept, we consider the simplest way to incorporate supervision in Equation 1:

$$\max_{\substack{\phi,\theta \\ s.t. \\ \mathbb{E}_{\mathbf{x},\mathbf{z}} R_s(q_\phi(\mathbf{z}|\mathbf{x}),\mathbf{z})\leq\kappa}} \mathrm{ELBO}(\phi,\theta) + \beta\mathbb{E}_{\mathbf{x}} R_u(q_\phi(\mathbf{z}|\mathbf{x})) \tag{2}$$

where $R_s(q_\phi(\mathbf{z}|\mathbf{x}),\mathbf{z})$ is some function computed only on the observed labels. In other words, we constrain the otherwise unsupervised problem using some supervised penalty. We now can include $R_s$ in the optimization as a regularizer under the KKT conditions:

$$\max_{\phi,\theta} \mathrm{ELBO}(\phi,\theta) + \beta\mathbb{E}_{\mathbf{x}} R_u(q_\phi(\mathbf{z}|\mathbf{x})) + \lambda\mathbb{E}_{\mathbf{x},\mathbf{z}} R_s(q_\phi(\mathbf{z}|\mathbf{x}),\mathbf{z}) \tag{3}$$

In our study, we use for $R_s$ a cross entropy loss on the first few dimension of the encoder's mean to align the representation to the factors of variation (which are normalized between zero and one during training and validation). When $\mathbf{z}$ has more dimension than the number of factors of variation the model is misspecified and we should learn to ignore the extra dimensions. As opposed to (Kingma et al., 2014), we directly observe the values of $\mathbf{z}$ and have no distinction between latent variables and class labels. (Narayanaswamy et al., 2017) introduce a semi-supervised generative model that disentangle some labelled factors of variation from the others that remain entangled. (Cheung et al., 2014), enrich the autoencoder objective with a supervised cross entropy loss that disentangle the label information (in their case the MNIST class information) from the other factors of variations (i.e. the digit's style) that can not be disentangled. Note that there is a rich literature on semi-supervised disentanglement, which unfortunately precedes the advent of the modern disentanglement metrics and problem formulation. Therefore, semi-supervised methods were only evaluated qualitatively and the quantitative gap between semi-supervised methods and unsupervised ones is currently not known. In the following section, we show that this simple model already competes with the SoTA approaches.

## 3 EXPERIMENTS

We build our experimental protocol on top of the `disentanglement_lib` of Locatello et al. (2018). For testing, we use the pipeline of `disentanglement_lib` which computes the disentanglement scores on 10000 labelled points.

**Setup for unsupervised methods:** We use the available labels (100 or 1000) to re-evaluate the trained models of Locatello et al. (2018) using MIG, SAP Score and DCI Disentanglement. These

results are then used to perform model selection on their trained models. We assume we can obtain exact labels. Note that the BetaVAE and FactorVAE metrics cannot be used for this purpose as they assume access to the generative process. We do not use Modularity to tune hyperparameters of the unsupervised models as it was not found to be strongly correlated with the other metrics in (Locatello et al., 2018) (see also Figure 1).

**Setup for semi-supervised method:** We add the supervised loss with its regularization strength to the $\beta$-VAE implementation of Locatello et al. (2018). For the supervised regularization strength we use the same hyperparameter range of the unsupervised one. We train more than 7000 models on the same seven datasets, with the same hyperparameters for 5 different random seeds. In this way, we can compare directly with their $> 12000$ SoTA unsupervised methods. We consider a clear train, validation and test split. We assume to have available either 100 or 1000 labelled datapoints which we use for training and model selection (90%/10% split). To test robustness, we consider exact labels (referred as *S2-perfect*), noisy labels (the observation of each factor of variation has a 10% chance of being random, referred as *S2-noisy*) and simplified labels (binned to only take 5 values, referred as *S2-bin*). This last scenario is meant to simulate the process of a human labelling a realistic number of images (either 100 or 1000) where each factor of variation can take only 5 values for simplicity. During training and validation, we group the values of each factor of variation in one dimension and normalize it so it takes values in [0,1]. This simple choice can be good for continuous variables (for example for spatial location) but is likely not ideal for discrete ones (such as shape). For model selection, we compute the supervised loss on the validation data.

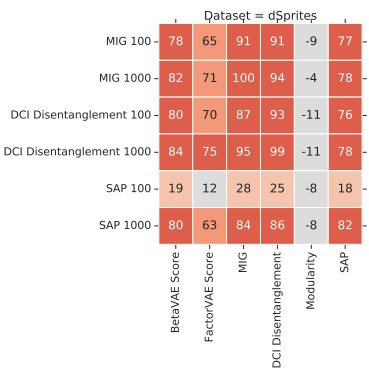

Figure 1: Rank correlation between metrics with 100 and 1000 samples versus 10 000 samples on dSprites.

| | Improvement |
|---|---|
| S2-perfect 100 | 82.2% |
| S2-perfect 1000 | 84.9% |
| S2-bin 100 | 79.1% |
| S2-bin 1000 | 84.9% |
| S2-noisy 100 | 75.6% |
| S2-noisy 1000 | 84.0% |

Table 1: How often the supervised loss improves upon the corresponding $\beta$-VAE with the same hyperparameters.

**Disentanglement metrics with few samples:** As a first step, we investigate whether using supervision to tune the hyperparameters of unsupervised methods is possible in the setting where only few labelled points are available. We re-evaluated the trained methods of (Locatello et al., 2018) with 100 and 1000 samples instead of 10 000 on a different set of datapoints. In Figure 1, we show the rank correlation of the MIG, DCI Disentanglement and SAP score on dSprites we use for validation compared to the test disentanglement scores. We notice a strong and consistent correlation. The only exception seem to be the SAP score with 100 samples that does not correlate strongly with any metric. These trends are consistent across different datasets.

From this comparison we conclude that it is possible to use a few labels to tune unsupervised models. Therefore, we argue that it is fair to compare model selection of unsupervised methods with semi-supervised approaches.

**Does supervision improve training?** In this section, we question whether we should keep all the labels to tune the hyperparameters or we should use some for training. For each dataset and metric, we take the $\beta$-VAE setup from Locatello et al. (2018) and compare their trained models with the same models trained with the additional supervised loss (only differences are seeds and supervised loss). Given the large amount of comparisons, we report how often does supervision improves upon the corresponding unsupervised method for each dataset and metric. In Table 1, we see that overall adding some supervision improves the unsupervised $\beta$-VAE, even if the supervision is noisy or not detailed (we use only 90% of the labels for training). Note that the result of Table 1 only tells us that good models exist, but not how to find them. Table 2, shows the comparison after model selection. For robust comparison, we group the models in 5 bins based on the seed and perform the selection over each bin, for each dataset and for each metric. From this experiment, we conclude that including labels during training appears to give better disentanglement than only performing supervised selection.

|  | S2-bin 100 | S2-noisy 100 | S2-perfect 100 |
|---|---|---|---|
| DCI Disentanglement 100 | 59.5% | 57.8% | 67.0% |
| MIG 100 | 63.2% | 60.5% | 67.0% |
| SAP 100 | 88.1% | 82.5% | 89.3% |

Table 2: How often the semi-supervised loss improves upon $\beta$-VAE after model selection using 100 labelled points.

|  | $\beta$-TCVAE | $\beta$-VAE | A-VAE | DIP-VAE-I | DIP-VAE-II | FactorVAE | S2-perfect |
|---|---|---|---|---|---|---|---|
| DCI | 18.6% | 13.6% | 9.7% | 17.1% | 3.9% | 7.4% | 29.8% |
| MIG | 23.5% | 10.4% | 8.5% | 16.9% | 5.4% | 7.7% | 27.7% |
| SAP | 9.0% | 6.4% | 9.9% | 7.3% | 6.4% | 7.3% | 53.6% |

Table 3: How often each method beats all the others after model selection using 100 labelled points. The semi-supervised method is most often the best one. DCI Disentanglement is abbreviated to DCI and AnnealedVAE to A-VAE, S2-perfect 100 to S2-perfect.

In Table 3, we show that using only 100 labels our semi-supervised $\beta$-VAE is competitive with all the other SoTA unsupervised models with supervised model selection. With 1000 labelled examples the gap is bigger as depicted in Table 5 in the appendix. The median score of the models tuned with the MIG with 100 labels versus our semi-supervised model can be seen in Tables 6-12 in the appendix. Results for all the metrics and datasets with confidence intervals can be seen in Figures 5-10. In Figure 11 we can see an example of the semi-supervised model perfectly disentangling Shapes3D. Overall, we conclude that the gain of using labels while training is higher with 1000 labels rather than 100. This implies that supervised model selection of the unsupervised models might be more useful when even fewer labels are available. On the other hand, if we have access to a considerable amount of labels, it seems better to use them during training.

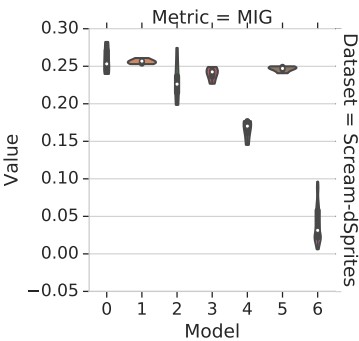

**Noisy labels:** Finally, we test how sensitive our semi-supervised method is to noisy or simplified labels. In Figure 2, we can see a standard behaviour where the variance significantly decreases when we add more labels. Collecting perfect labels seem to be the best strategy, but label noise and the simplified labels do not significantly harm the performances. The full result can be seen in Figure 4 in the Appendix. From this experiment, we conclude that it might be beneficial to collect some labels for training even if those labels are not exactly accurate.

## 4 CONCLUSIONS

Figure 2: Comparing different labelling strategies. Models are abbreviated as (0=S2-perfect 100, 1=S2-perfect 1000, 2=S2-bin 100, 3=S2-bin 1000, 4=S2-noisy 100, 5=S2-noisy 1000, 6=VAE)

In this paper, we investigate how a practitioner interested in learning disentangled representation on some dataset could approach the problem. We empirically observed that a very limited amount of supervision allows for supervised model selection in the setup of Locatello et al. (2018). We also tested whether with the same amount of supervision, one could obtain better disentanglement if the labels were used for training and observed that it is often the case. While unsupervised model selection remains a big open challenge for the community, we argue that disentanglement methods could be reliably used in practice as long as a little information about the factors of variation is available. With this paper, we hope to renew the interest in semi-supervised disentanglement learning which is nowadays less popular than unsupervised training with supervised model selection. Notably, the impossibility result of Locatello et al. (2018) does not apply in this setting and our experiments show promising results. As future work, we aim at testing model selection using the disentanglement metrics with binned labels and perform a large scale study comparing the effect of the supervised loss on other unsupervised methods.

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

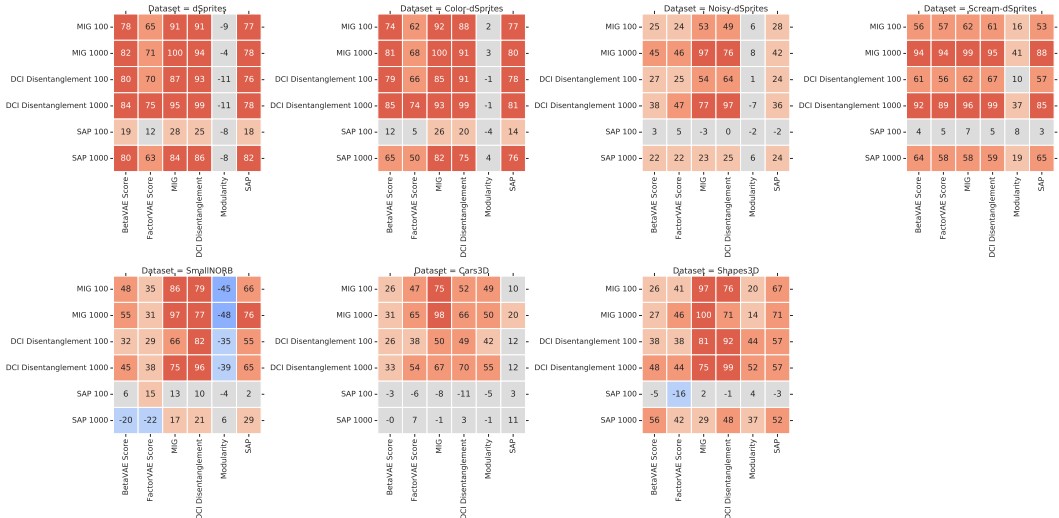

Figure 3: Rank correlation between metrics with 100 and 1000 samples versus 10 000 samples on all the datasets.

## A    ARCHITECTURE AND SETUP

We build these experiments using the `disentanglement_lib` of Locatello et al. (2018). We add a cross entropy loss to the first few dimensions of the mean representation of the encoder. We test the following values for the hyperparameter in front of the supervised loss: [1, 2, 4, 6, 8, 16]. These values are the same as the unsupervised regularizer. In all our experiments, we use either 100 or 1000 labels. In the semi supervised models, we used 90% of the labels to train and the rest for validation. We consider the cases where labels are perfect, noisy (10% are random) or binned in five values. The disentanglement metrics for testing always use 10 000 labels for numerical stability. We run the semi-supervised training for 5 different random seeds. Optimizer, architectures, latent space size, batch size are all identical to the ones used in Locatello et al. (2018).

## B    ADDITIONAL RESULTS

In Figure 3 we show the rank correlation between the different disesntanglement metrics and the ones we use to tune hyperparameters. We can clearly see a consistent correlation (except with SAP 100) between the metrics across the different datasets. We conclude that using these supervised metrics to tune hyperparameters of unsupervised models is possible even if only few labels are available. In table 4, we present how often the simple semi-supervised method beats supervised model selection using 1000 samples. We can see that the gap between the two approaches is bigger than with 100 samples as we are able to incorporate more supervision while training. In table 5, we see a similar trend, this time comparing against all the unsupervised methods and not only $\beta$-VAE. In tables 6-12 we present the median values of the disentanglement metrics achieved after model selection using the validation loss for the semi-supervised model with perfect labels and the MIG. We assume to have available 100 samples to compute the MIG and 10 for the validation loss as the other 90 are used for training. In Figures 5-10, we show the confidence bound for these results. In Figure 4, we can see that the semi-supervised model is relatively robust to noisy and simplified labels.

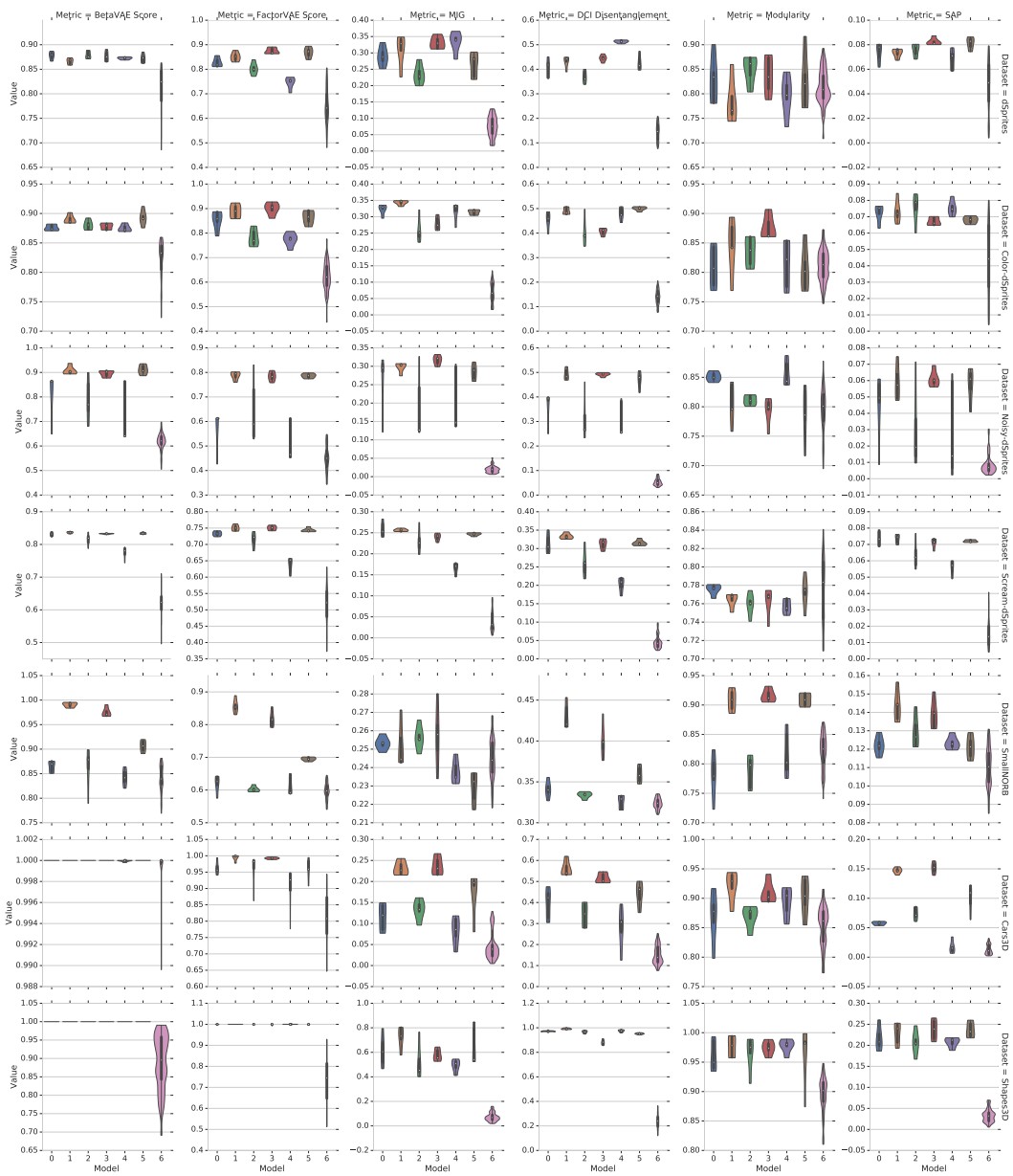

Figure 4: Comparing different labelling strategies. Models are abbreviated as (0=S2-perfect 100, 1=S2-perfect 1000, 2=S2-bin 100, 3=S2-bin 1000, 4=S2-noisy 100, 5=S2-noisy 1000, 6=VAE)

|  | S2-bin 1000 | S2-noisy 1000 | S2-perfect 1000 |
| --- | --- | --- | --- |
| DCI Disentanglement 1000 | 73.8% | 69.4% | 79.5% |
| MIG 1000 | 72.4% | 69.1% | 78.3% |
| SAP 1000 | 82.3% | 75.9% | 85.0% |

Table 4: How often the semi supervised loss improves upon $\beta$-VAE after model selection using 1000 labelled points.

|  | $\beta$-TCVAE | $\beta$-VAE | A-VAE | DIP-VAE-I | DIP-VAE-II | FactorVAE | S2-perfect 1000 |
|---|---|---|---|---|---|---|---|
| DCI | 14.0% | 7.6% | 8.7% | 9.8% | 3.4% | 7.2% | 49.2% |
| MIG | 14.2% | 8.8% | 8.1% | 9.2% | 5.8% | 5.4% | 48.5% |
| SAP | 10.7% | 7.3% | 7.7% | 10.3% | 4.2% | 4.6% | 55.2% |

Table 5: How often each method beats all the others after model selection using 1000 labelled points. The gap between the semi-supervised method and supervised model selection is larger than with 100 labels. DCI Disentanglement is abbreviated to DCI and AnnealedVAE to A-VAE.

| Model | BetaVAE Score | DCI | FactorVAE Score | MIG | Modularity | SAP |
|---|---|---|---|---|---|---|
| $\beta$-TCVAE | 0.880 | 0.465 | 0.821 | 0.323 | 0.842 | 0.059 |
| $\beta$-VAE | 0.873 | 0.514 | 0.712 | 0.354 | 0.795 | 0.048 |
| AnnealedVAE | 0.866 | 0.397 | 0.595 | 0.363 | 0.781 | 0.070 |
| DIP-VAE-I | 0.852 | 0.193 | 0.606 | 0.144 | 0.921 | 0.076 |
| DIP-VAE-II | 0.851 | 0.169 | 0.602 | 0.105 | 0.868 | 0.064 |
| FactorVAE | 0.865 | 0.405 | 0.817 | 0.292 | 0.827 | 0.063 |
| S2-perfect 100 | 0.871 | 0.516 | 0.752 | 0.300 | 0.794 | 0.060 |

Table 6: Median scores on dSprites with 100 labels. Unsupervised methods tuned with MIG

| Model | BetaVAE Score | DCI | FactorVAE Score | MIG | Modularity | SAP |
|---|---|---|---|---|---|---|
| $\beta$-TCVAE | 0.890 | 0.491 | 0.853 | 0.269 | 0.901 | 0.068 |
| $\beta$-VAE | 0.870 | 0.387 | 0.747 | 0.243 | 0.791 | 0.077 |
| AnnealedVAE | 0.870 | 0.438 | 0.607 | 0.376 | 0.812 | 0.077 |
| DIP-VAE-I | 0.861 | 0.193 | 0.622 | 0.133 | 0.914 | 0.074 |
| DIP-VAE-II | 0.843 | 0.164 | 0.588 | 0.095 | 0.850 | 0.062 |
| FactorVAE | 0.891 | 0.428 | 0.811 | 0.341 | 0.822 | 0.065 |
| S2-perfect 100 | 0.870 | 0.432 | 0.862 | 0.308 | 0.849 | 0.075 |

Table 7: Median scores on Color-dSprites with 100 labels. Unsupervised methods tuned with MIG

| Model | BetaVAE Score | DCI | FactorVAE Score | MIG | Modularity | SAP |
|---|---|---|---|---|---|---|
| $\beta$-TCVAE | 0.712 | 0.176 | 0.448 | 0.095 | 0.843 | 0.008 |
| $\beta$-VAE | 0.649 | 0.237 | 0.452 | 0.129 | 0.862 | 0.006 |
| AnnealedVAE | 0.812 | 0.185 | 0.600 | 0.206 | 0.871 | 0.047 |
| DIP-VAE-I | 0.832 | 0.142 | 0.613 | 0.092 | 0.883 | 0.053 |
| DIP-VAE-II | 0.786 | 0.133 | 0.562 | 0.098 | 0.834 | 0.007 |
| FactorVAE | 0.769 | 0.188 | 0.645 | 0.105 | 0.759 | 0.016 |
| S2-perfect 100 | 0.866 | 0.394 | 0.612 | 0.304 | 0.843 | 0.062 |

Table 8: Median scores on Noisy-dSprites with 100 labels. Unsupervised methods tuned with MIG

| Model | BetaVAE Score | DCI | FactorVAE Score | MIG | Modularity | SAP |
|---|---|---|---|---|---|---|
| $\beta$-TCVAE | 0.823 | 0.270 | 0.728 | 0.245 | 0.755 | 0.066 |
| $\beta$-VAE | 0.820 | 0.264 | 0.726 | 0.224 | 0.761 | 0.055 |
| AnnealedVAE | 0.200 | 0.004 | 0.215 | 0.001 | 0.641 | 0.002 |
| DIP-VAE-I | 0.656 | 0.136 | 0.563 | 0.138 | 0.738 | 0.024 |
| DIP-VAE-II | 0.727 | 0.158 | 0.634 | 0.141 | 0.742 | 0.040 |
| FactorVAE | 0.740 | 0.154 | 0.587 | 0.131 | 0.736 | 0.048 |
| S2-perfect 100 | 0.834 | 0.336 | 0.736 | 0.266 | 0.769 | 0.074 |

Table 9: Median scores on Scream-dSprites with 100 labels. Unsupervised methods tuned with MIG

| Model | BetaVAE Score | DCI | FactorVAE Score | MIG | Modularity | SAP |
|---|---|---|---|---|---|---|
| $\beta$-TCVAE | 0.850 | 0.335 | 0.575 | 0.245 | 0.862 | 0.099 |
| $\beta$-VAE | 0.833 | 0.318 | 0.552 | 0.242 | 0.905 | 0.098 |
| AnnealedVAE | 0.555 | 0.106 | 0.325 | 0.108 | 0.981 | 0.070 |
| DIP-VAE-I | 0.882 | 0.294 | 0.704 | 0.268 | 0.794 | 0.111 |
| DIP-VAE-II | 0.862 | 0.296 | 0.569 | 0.255 | 0.860 | 0.123 |
| FactorVAE | 0.689 | 0.328 | 0.629 | 0.261 | 0.818 | 0.107 |
| S2-perfect 100 | 0.855 | 0.330 | 0.627 | 0.239 | 0.826 | 0.112 |

Table 10: Median scores on SmallNORB with 100 labels. Unsupervised methods tuned with MIG

| Model | BetaVAE Score | DCI | FactorVAE Score | MIG | Modularity | SAP |
|---|---|---|---|---|---|---|
| $\beta$-TCVAE | 1.000 | 0.482 | 0.893 | 0.206 | 0.934 | 0.016 |
| $\beta$-VAE | 1.000 | 0.396 | 0.883 | 0.111 | 0.911 | 0.007 |
| AnnealedVAE | 1.000 | 0.143 | 0.824 | 0.079 | 0.842 | 0.009 |
| DIP-VAE-I | 1.000 | 0.245 | 0.945 | 0.096 | 0.831 | 0.041 |
| DIP-VAE-II | 1.000 | 0.260 | 0.932 | 0.098 | 0.857 | 0.010 |
| FactorVAE | 1.000 | 0.275 | 0.941 | 0.119 | 0.928 | 0.016 |
| S2-perfect 100 | 1.000 | 0.385 | 0.954 | 0.119 | 0.925 | 0.040 |

Table 11: Median scores on Cars3D with 100 labels. Unsupervised methods tuned with MIG

| Model | BetaVAE Score | DCI | FactorVAE Score | MIG | Modularity | SAP |
|---|---|---|---|---|---|---|
| $\beta$-TCVAE | 1.000 | 0.960 | 1.000 | 0.696 | 0.958 | 0.183 |
| $\beta$-VAE | 1.000 | 0.849 | 0.963 | 0.722 | 0.868 | 0.166 |
| AnnealedVAE | 0.942 | 0.697 | 0.906 | 0.545 | 0.896 | 0.066 |
| DIP-VAE-I | 1.000 | 0.820 | 0.902 | 0.336 | 0.973 | 0.115 |
| DIP-VAE-II | 0.984 | 0.591 | 0.958 | 0.466 | 0.896 | 0.100 |
| FactorVAE | 1.000 | 0.793 | 0.952 | 0.650 | 0.923 | 0.091 |
| S2-perfect 100 | 1.000 | 0.934 | 1.000 | 0.530 | 0.977 | 0.193 |

Table 12: Median scores on Shapes3D with 100 labels. Unsupervised methods tuned with MIG

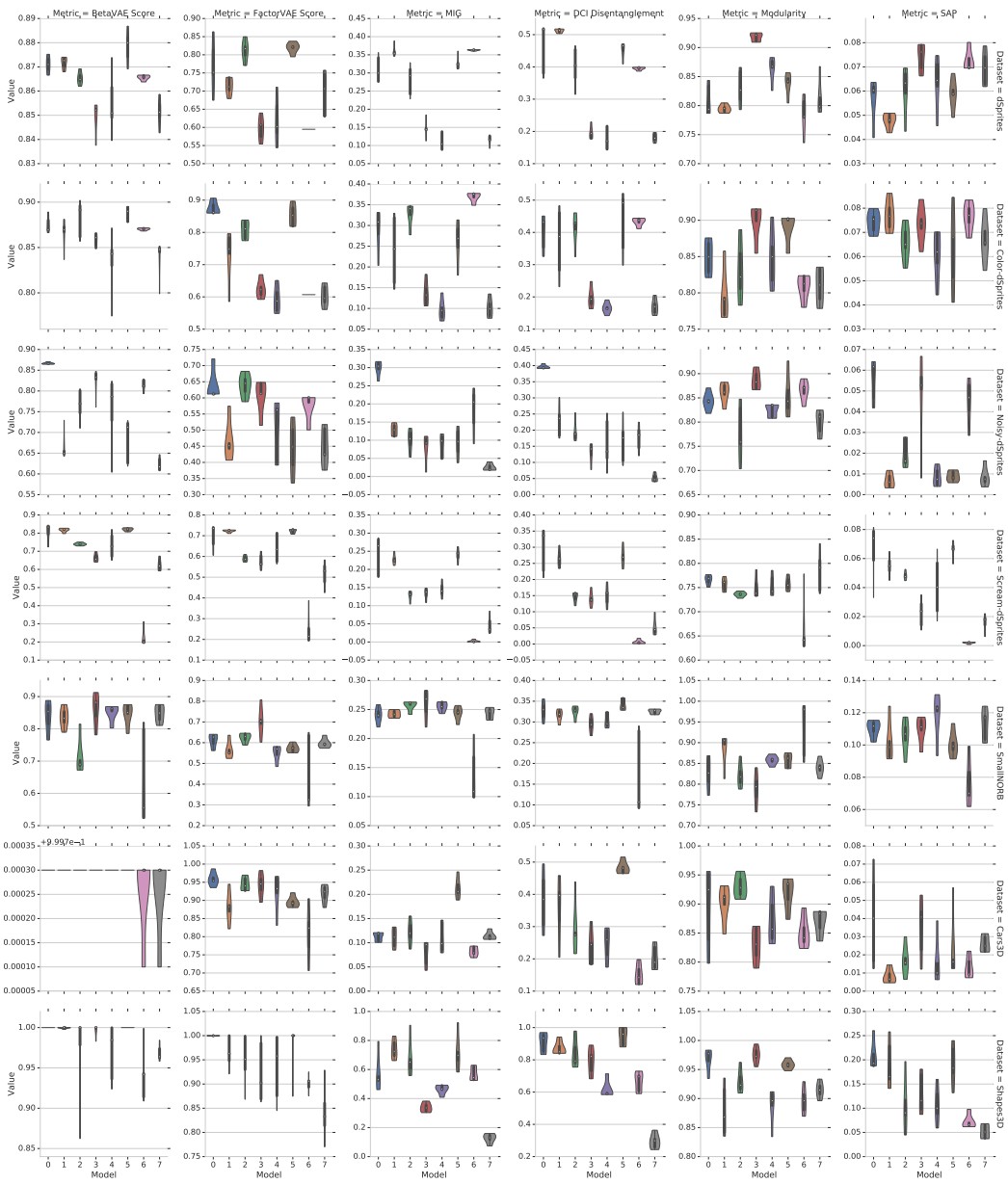

Figure 5: Confidence intervals for the models with 100 labels. Unsupervised models are tuned using the MIG. Models are abbreviated as (0=S2-perfect 100, 1=$\beta$-VAE, 2=FactorVAE, 3=DIP-VAE-I, 4=DIP-VAE-II, 5=$\beta$-TCVAE, 6=AnnealedVAE, 7=VAE).

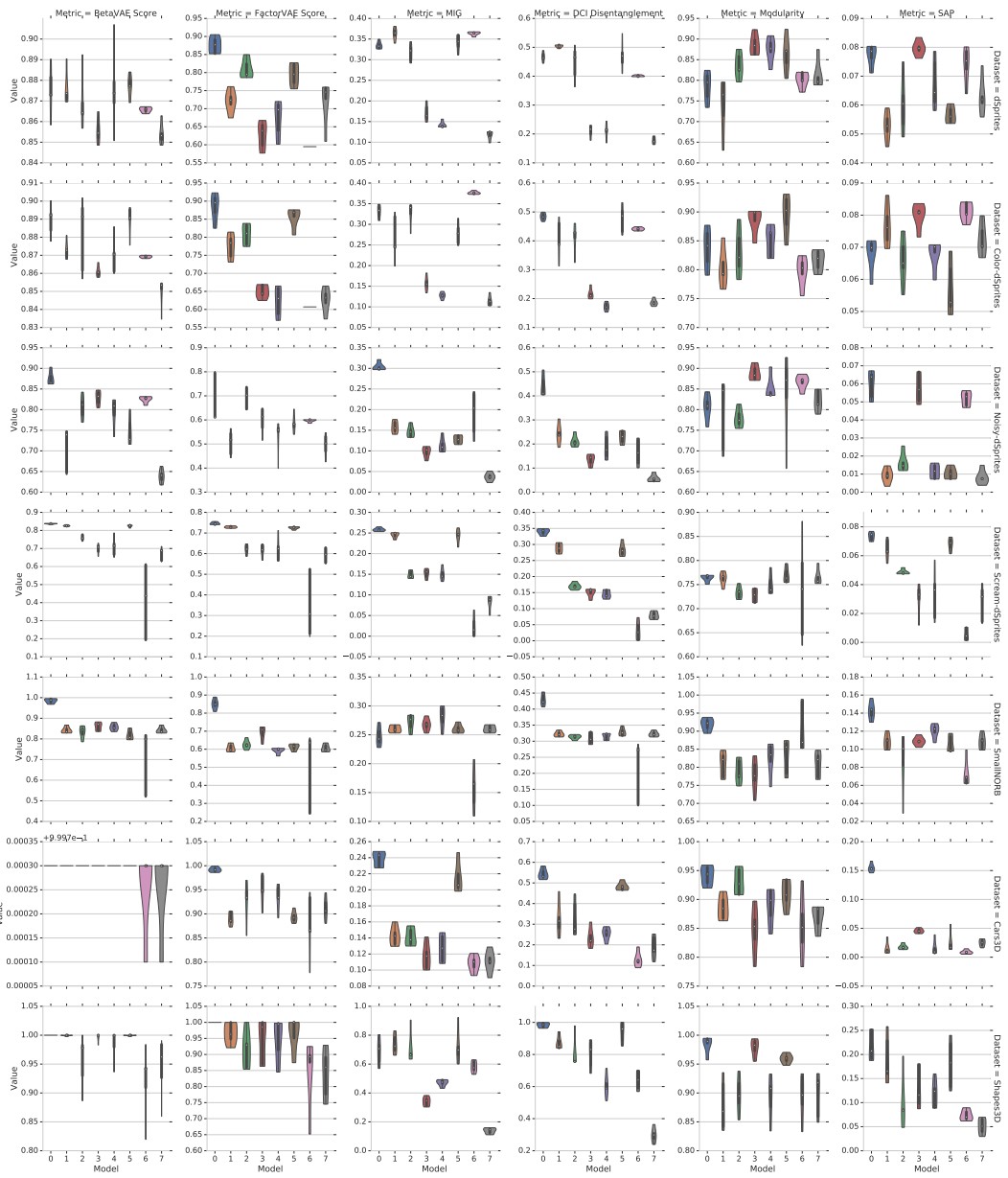

Figure 6: Confidence intervals for the models with 1000 labels. Unsupervised models are tuned using the MIG. Models are abbreviated as (0=S2-perfect 1000, 1=$\beta$-VAE, 2=FactorVAE, 3=DIP-VAE-I, 4=DIP-VAE-II, 5=$\beta$-TCVAE, 6=AnnealedVAE, 7=VAE).

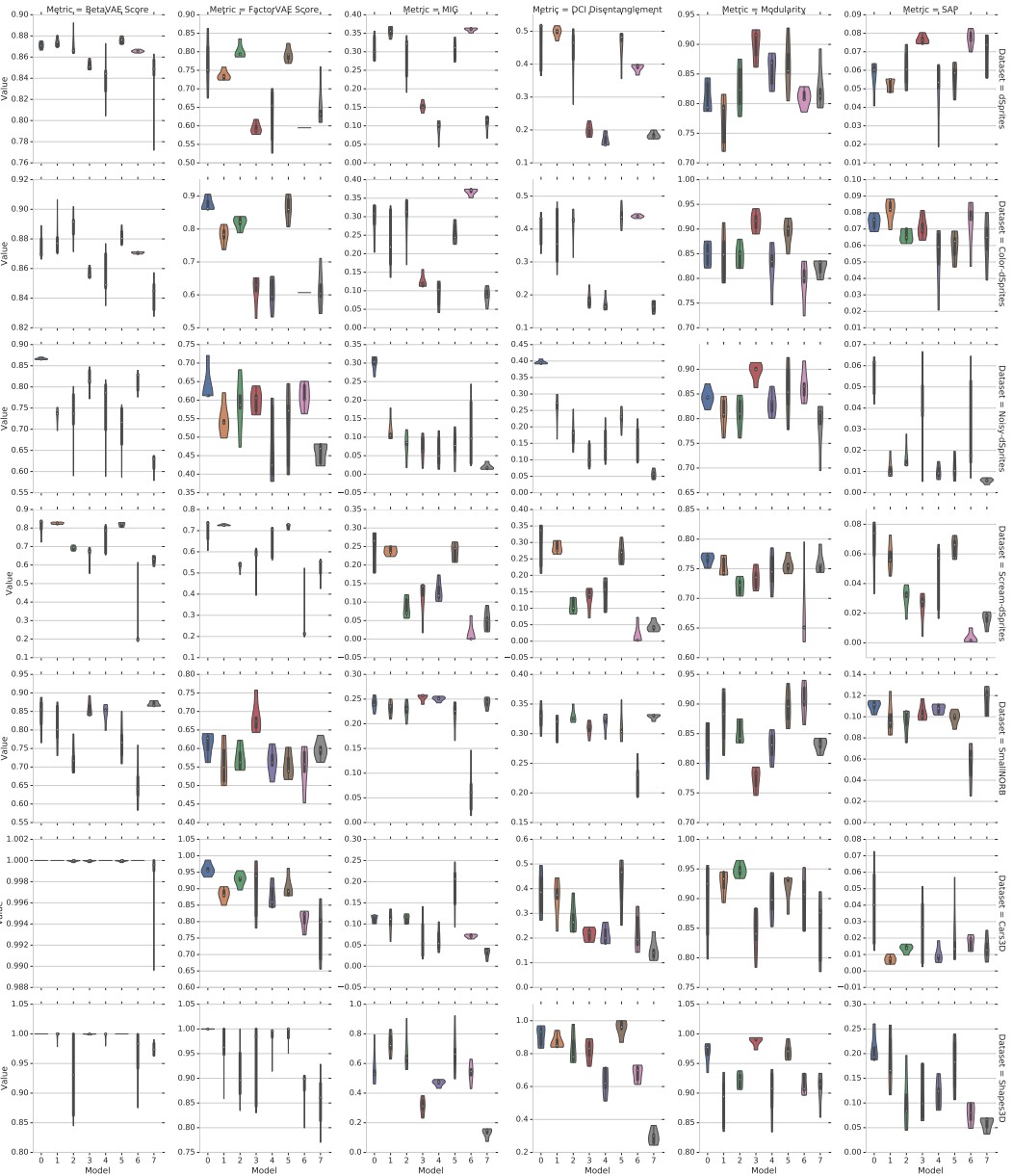

Figure 7: Confidence intervals for the models with 100 labels. Unsupervised models are tuned using the DCI Disentanglement. Models are abbreviated as (0=S2-perfect 100, 1=$\beta$-VAE, 2=FactorVAE, 3=DIP-VAE-I, 4=DIP-VAE-II, 5=$\beta$-TCVAE, 6=AnnealedVAE, 7=VAE).

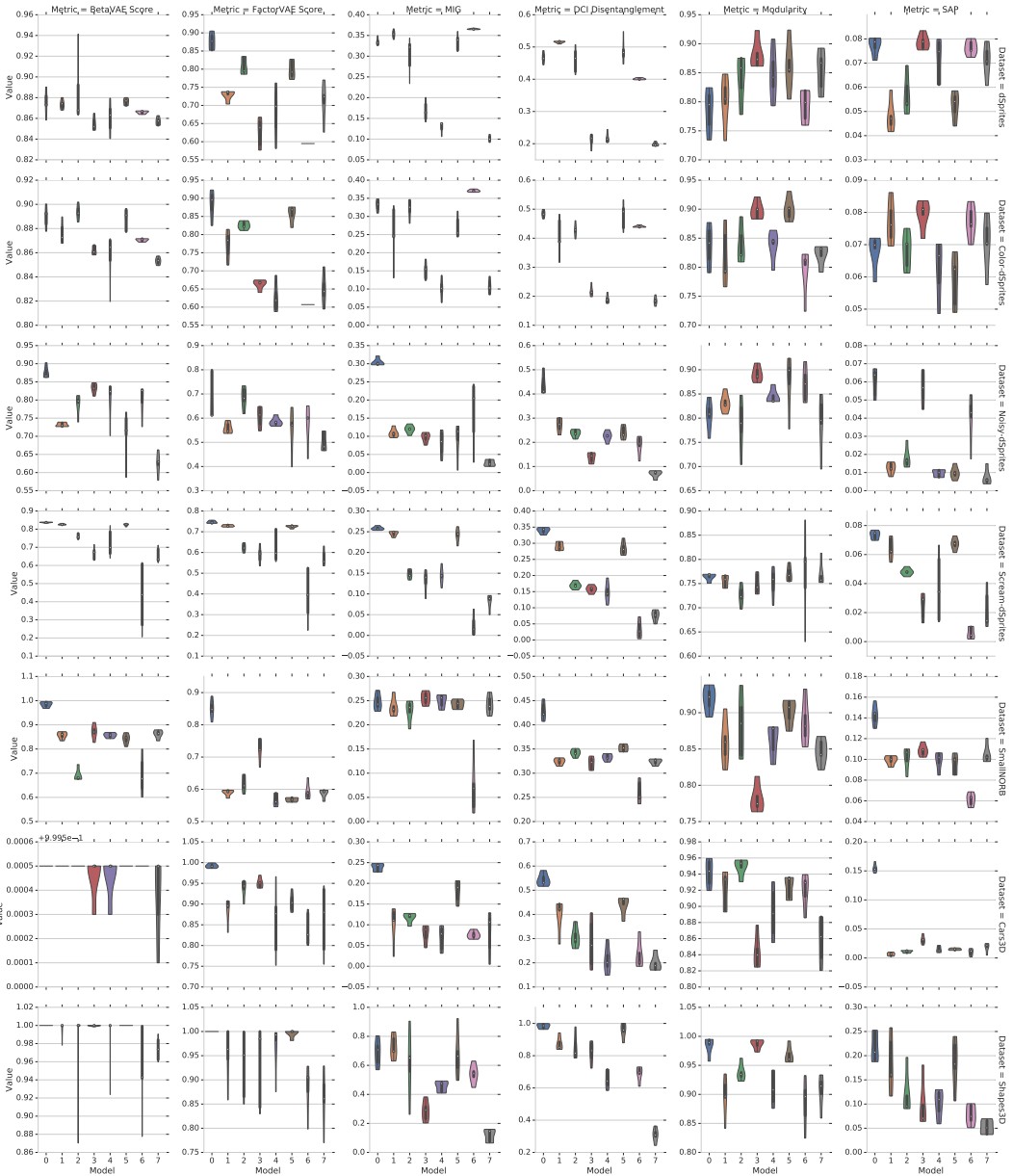

Figure 8: Confidence intervals for the models with 1000 labels. Unsupervised models are tuned using the DCI Disentanglement. Models are abbreviated as (0=S2-perfect 1000, 1=$\beta$-VAE, 2=FactorVAE, 3=DIP-VAE-I, 4=DIP-VAE-II, 5=$\beta$-TCVAE, 6=AnnealedVAE, 7=VAE).

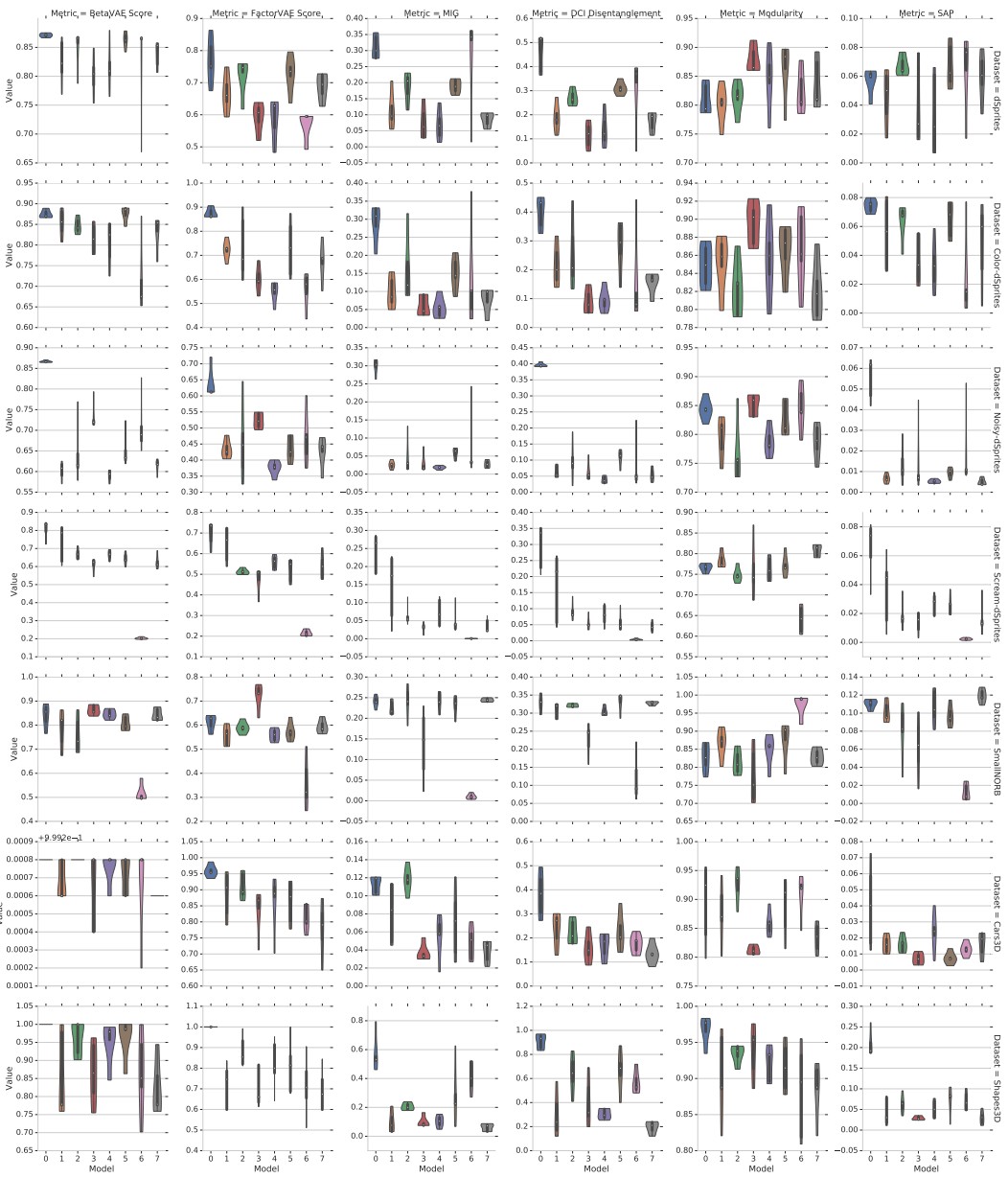

Figure 9: Confidence intervals for the models with 100 labels. Unsupervised models are tuned using the SAP Score. Models are abbreviated as (0=S2-perfect 100, 1=$\beta$-VAE, 2=FactorVAE, 3=DIP-VAE-I, 4=DIP-VAE-II, 5=$\beta$-TCVAE, 6=AnnealedVAE, 7=VAE).

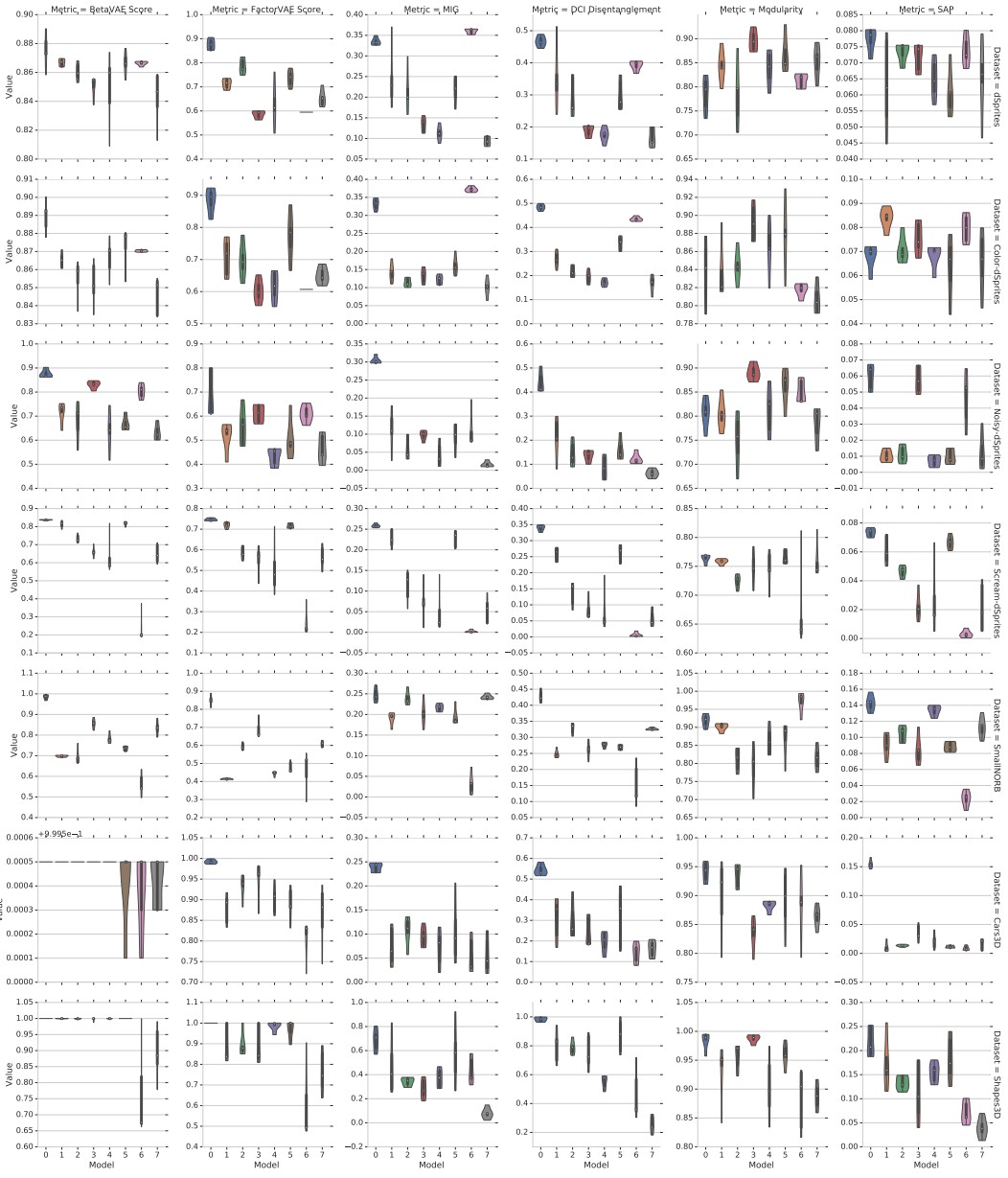

Figure 10: Confidence intervals for the models with 1000 labels. Unsupervised models are tuned using the SAP Score. Models are abbreviated as (0=S2-perfect 1000, 1=$\beta$-VAE, 2=FactorVAE, 3=DIP-VAE-I, 4=DIP-VAE-II, 5=$\beta$-TCVAE, 6=AnnealedVAE, 7=VAE).

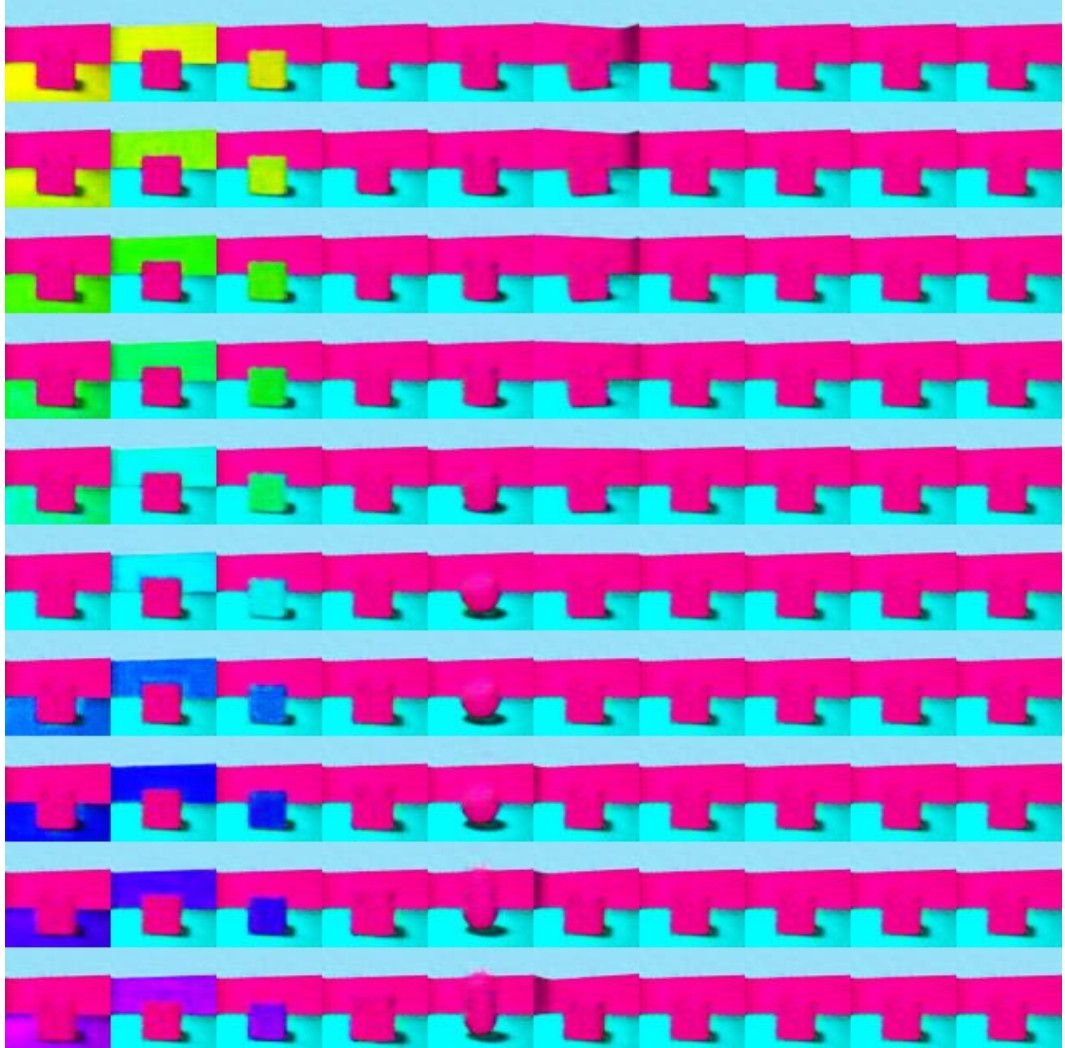

Figure 11: Example of semi-supervised model with 100 labels achieving perfect disentanglement on Shapes3D. The model correctly learns to ignore the last four channels of the representation.

