# OpenReview forum: "Disentangling Factors of Variations Using Few Labels"
_ICLR.cc/2019/Workshop/LLD — LLD 2019_

### Official Review · AnonReviewer1 · 2019-04-07
**Interesting observation and proof-of-concept**

**Rating:** 4
**Confidence:** 2

**Review:**

The authors of this work consider the problem of learning disentangled representations. They observe that if labeled data are used for model selection, than using them for training can lead to interesting results. They propose to regularize beta-VAE with a cross entropy loss for a small portion of labeled data, and show improvement on the models that use labeled data only for model selection. The experimental results indeed validate the authors hypothesis and show that even a small set of partially of noisily labeled data can have an interesting impact on the training.

The observation and initial results are interesting, and the work is wort to be discussed in this workshop.

---

### Official Review · AnonReviewer2 · 2019-04-08
**Semisupervised learning by introducing observed labels in unsupervised learning methods yields non trivial results for few labels**

**Rating:** 3
**Confidence:** 2

**Review:**

The papers propose a model in which some of the latent variables of an unsupervised model are associated with a small number of user defined labels. This increases the discriminativity of the model and enable it to separate factors of variation associated with those labels.
The introduction of the new term in the objective essentially places the model withing the domain of supervised models (or at least semisupervised). However, all the numerical test appear to be against unsupervised models. Improvement by those standards seems trivial to me. I would urge the authors to include comparisons against other semisupervised methods that work on similar numbers of labeled data

---

### Decision · Program_Chairs · 2019-04-16
**Acceptance Decision**

Accept